# Fabrication of ZnO-Al_2_O_3_-PTFE Multilayer Nano-Structured Functional Film on Cellulose Insulation Polymer Surface and Its Effect on Moisture Inhibition and Dielectric Properties

**DOI:** 10.3390/polym11081367

**Published:** 2019-08-19

**Authors:** Cong Liu, Jian Hao, Yanqing Li, Ruijin Liao

**Affiliations:** State Key Laboratory of Power Transmission Equipment & System Security and New Technology, Chongqing University, Chongqing 400044, China

**Keywords:** cellulose insulation polymer, nano-structured functional film, magnetron sputtering, hydrophobicity, dielectric properties

## Abstract

After a century of practice, cellulose insulating polymer (insulating paper/pressboard) has been shown to be one of the best and most widely used insulating materials in power transformers. However, with the increased voltage level of the transformer, research has focused on improving the insulation performance of the transformer’s cellulose insulation polymer. Considering the complex environment of the transformer, it is not enough to improve the single performance of the insulating polymer. In this study, a nano-structured ZnO-Al_2_O_3_-PTFE (polytetrafluoroethylene) multifunctional film was deposited on the surface of insulating pressboard by radio frequency (RF) magnetron sputtering. The effect of the multilayered ZnO-Al_2_O_3_-PTFE functional film on the dielectric and water contact angle of the cellulose insulating polymer was investigated. The scanning electron microscopy/energy dispersive spectrometry (SEM/EDS) showed that the nano-structured ZnO-Al_2_O_3_-PTFE functional film was successfully deposited on the cellulose insulation pressboard surface. The functional film presented an obvious stratification phenomenon. By analyzing the result of the contact angle, it was found that the functional film shields the hydroxyl group of the inner cellulose and improves hydrophobicity. The AC breakdown field strength of the treated samples was obviously increased (by 12 to ~17%), which means that the modified samples had a better dielectric insulation performance. This study provides a surface modification method to comprehensively improve electrical properties and the ability to inhibit the moisture of the cellulose insulating polymer, used in a power transformer.

## 1. Introduction

After a century of practice, cellulose insulating polymer (insulating paper/pressboard) has been shown to be one of the best and most widely used insulating materials in transformers [1,2,3]. For cellulose insulation polymer-based transformers, the cellulose insulation polymer itself has poor heat dissipation and the air gap in the fiber reduces its dielectric strength, so the cellulose insulating polymer needs to be immersed in the insulating oil. The insulation oil fills the air gap and has functions in heat dissipation, sealing, and insulation coordination [1]. In addition, the insulation oil can solve the problem of aging through oil purification, regeneration treatment, or even new oil, but for the insulation paper, the deterioration caused by the decline in performance is irreversible [4]. Therefore, the insulating property of the paper/pressboard is a key factor to the whole insulation system of most power transformers in long-term operations [5].

There are many factors leading to transformer insulation failure. Among them, the electric field, temperature, and moisture play dominant roles in the aging process of transformer insulation paper, and these factors also have a synergistic effect on the aging of insulating paper [4,5,6,7]. Under the action of the thermal field, insulating paper will gradually age and the molecular chains of the cellulose will break. Additionally, moisture can directly participate in the degradation of the cellulose and accelerate the process. In cellulose, each glucose molecule contains three hydroxyl groups with strong hydrophilicity. Once water molecules are encountered, free hydroxyl groups will form hydrogen bonds with the adsorbed water molecules. The more water adsorbed in the insulating paper, the faster the hydrolysis rate of the cellulose [7,8], and the faster the decrease in the insulation performance of the insulating paper [9,10,11]. When the insulation performance of insulating paper is reduced to the threshold value, dielectric breakdown will occur and the insulation performance becomes completely lost [5]. Therefore, in the oil-immersed transformer, reducing the moisture content of the insulating paper and improving the electrical properties of the insulating paper is very important for improving the service life of the insulating paper [6,7].

In order to reduce the hydrophilicity of cellulose molecules, other more stable chemical groups (such as cyanoethyl and acetic anhydride) have replaced polar hydroxyl groups in cellulose molecules for a long time [12,13]. However, the cellulose chain between the hydrogen bond, is destroyed because of the reduction of the hydroxyl group, leading to a decrease in the overall mechanical strength of the insulation paper. By contrast, the physical modification method will not destroy the mechanical properties of insulating paper. When amine compounds were added to the insulating paper, they improved its properties by consuming the moisture produced in the aging process [13,14,15,16]. Doping of montmorillonite, SiO_2_ hollow microspheres, ZnO nanoparticles, and Al_2_O_3_ nanoparticles in insulating paper can effectively improve the dielectric properties of insulating paper [17,18,19,20,21]. The existing methods can improve the ability of insulating paper in some respects, but there is still a lack of a comprehensive method to improve the electrical properties of insulating paper and improve its ability to inhibit moisture.

In recent years, the wide band-gap semi-conductor material zinc oxide (ZnO) has attracted more and more attention [22], with advantages of high electron mobility and nonlinear resistance [22,23,24]. Nano-ZnO is believed to be beneficial to charge transport [24], and its resistance decreases with the increase of the electric field, which helps to balance the electric field [25]. Low concentration doped nano-ZnO, in low density polyethylene (LDPE) will be beneficial in reducing the internal electric field distortion and improves its breakdown performance [25,26]. Al_2_O_3_ is a famous insulating material, often used as a coating or nano-filler for insulating paper [27,28,29,30]. The results show that the nano-Al_2_O_3_ doped insulation paper possesses the better dielectric properties and AC breakdown strength [30]. In addition, the Al_2_O_3_ functional film, constructed on the surface of the insulating board, can realize binding charge and effectively inhibit charge injection from electrodes [31]. Polytetrafluoroethylene (PTFE) has good insulation properties, corrosion resistance, and hydrophobic properties. It prevents moisture in oil from entering the insulating pressboard [32,33,34,35] and, on the other hand, it can protect the functional layer of the inner layer from being damaged [33]. Additionally, PTFE film can be used as the outermost protective layer of a multilayer film.

In this study, a multifunctional composite film was deposited on the surface of insulating pressboard by magnetron sputtering. Firstly, the physical and chemical characteristics of the functional film were obtained. The influence of the functional, thin film on the ability to inhibit moisture and electrical properties were investigated. As a result, the functional, thin film successfully improved the electrical properties of the cellulose insulating polymer and its ability to inhibit moisture.

## 2. Materials and Methods

### 2.1. Materials and Sample Preparation

The samples used in the experiment were 0.5 mm thick commercial insulating pressboard, produced in the same batch, and the pressboard was cut to the size of 15 cm × 10 cm. For the preparation of the multilayer thin film, the JPGF-480 reactive radio frequency (RF) magnetron sputtering device at 13.56 MHz (Beijing Instrument Factory, Beijing, China) was used to form the ZnO, Al_2_O_3_, and PTFE films. The principle of magnetron sputtering is shown in [31,35,36]. First, the vacuum was pumped to 4 × 10^−3^ Pa through mechanical and molecular pumps. Argon gas (Chongqing Hong Hao Gas Co., Ltd., Chongqing, China) was used at a constant pressure of 1.5 Pa. Oxygen gas (Chongqing Hong Hao Gas Co., Ltd., Chongqing, China) was used at a flow of 20 sccm. The forward power was 100 W. For the deposition of ZnO film, a zinc target of 99.999% purity was sputtered in oxygen atmosphere for 10 min. For the deposition of Al_2_O_3_ film, an aluminum target of 99.999% purity was sputtered in oxygen atmosphere for 30 min and 60 min. The PTFE target was sputtered for 20 min and 30 min, without reaction oxygen gas, to prepare the PTFE films on the surface of the insulation pressboard. All targets were 61.5 mm in diameter and 6 mm in thickness.

The distance between the target and the substrate was about 10 cm, and the ambient temperature was 28 °C. The preparation process of the ZnO-Al_2_O_3_-PTFE film in this paper is shown in Figure 1. First, a ZnO film was coated on both sides of the insulating paperboard, then the Al_2_O_3_ films were coated and finally the PTFE films. For the convenience of the article description, the sample abbreviations are listed in Table 1. Among the samples, Z10-A30-P20 means that the ZnO film was first coated for 10 min, then the Al_2_O_3_ film was coated for 30 min, and finally the PTFE template was coated for 20 min.

### 2.2. Sample Processing and Characterization Methods

The process of sample preparation and characterization is shown in Figure 2. The treated and untreated samples were cut into the size of 5 × 5 mm. The samples and mineral oil were respectively placed in the vacuum box and dried for 48 h at 90 °C and 50 Pa to remove moisture. Then, the temperature of the vacuum box was adjusted to 60 °C. The oil and pressboards were placed in a jar with a vacuum of 50 Pa. The sample was left for 48 h to ensure full immersion. Finally, dry nitrogen, with a humidity less than 5 μL/L, was injected until the pressure inside the box was restored to atmospheric pressure; this ensured that the box was filled with dry nitrogen. Then, the jar was sealed under the nitrogen environment.

Different treated samples were used for different tests. The samples for the SEM/energy dispersive spectrometry (EDS) and contact angle test were pressboards without oil, while the samples for thermogravimetry (TG), frequency dielectric spectroscopy (FDS), and the breakdown test were pressboards after oil immersion treatment. The microstructure and composition of the sample were described by scanning electron microscopy/energy dispersive spectrometry (SEM/EDS, JSM-7800F, JEOL, Tokyo, Japan). The insulation pressboard is relatively soft, and a direct use of cutting will often destroy the insulation pressboard cross-section. It is therefore impossible to observe the original shape of the insulation cardboard cross-section. Therefore, in this paper, argon ion cross-section polishing (Gatan 697 llion II, Gatan, America) was used to obtain a cross-section of the sample. The principle of argon ion cross-section polishing is shown in Figure 3. Figure 3a is the schematic diagram of argon ion polishing, Figure 3b shows the effect of the sample section polishing, and Figure 3c shows the side view. As in the processing energy source, the argon ion beam hit perpendicular to the surface of a specimen, and the side of the sample was exposed to the ion beam with the help of a baffle. The argon ions generated bombarded the sample surface at a high speed. In under 1 h and 50 min at 6.5 keV of ion beam energy, 20 min at 3 keV, and 20 min at 1 keV, ions made a cross-section perpendicular to the surface of a specimen. The samples were stored in vacuum and then tested by SEM/EDS. The repellency of the samples to water and insulating oil was tested with a Kyowa contact angle meter at least three times per sample. TG was used to evaluate the immersion rate of the samples. Each sample weighed about 5 mg, and the TG and derivative thermogravimetry (DTG) curves were measured at a rate of 7 °C/min, and in a range of 30–550 °C under nitrogen atmosphere. Novo-control Concept 80 Broadband dielectric spectroscopy equipment (Novocontrol GmbH, Montabaur, Germany) was used to get FDS from 10^−1^ to 10^7^ Hz at room temperature. In addition, the dielectric constant and dielectric loss of different samples were analyzed by FDS to obtain polarization information of the samples. Finally, the breakdown performance of the sample was tested to characterize its insulation capability. The AC breakdown test adopts the equal-diameter electrode (diameter 25 mm) in the international standard IEC 60243. For the short-term test, the breakdown frequency was 50 Hz, the boosting speed was 1000 V/s, and each sample was subjected to six times at 25 °C. The test results are statistically based on the two-parameter Weibull distribution in the Chinese national standard GB/T 29310-2012.

## 3. Results and Discussions

### 3.1. SEM/EDS

#### 3.1.1. Surface Topography Analysis

The SEM micrographs of the untreated and deposited functional film insulating pressboard, without being ion polished, is shown in Figure 4. Figure 4a,b are SEM images for the new pressboard surface at 2000× and 20,000× magnification. As can be seen from Figure 4a, a bundle of fibers is made up of many cellulose fibers. Fiber bundles are interlaced with each other, presenting a layered structure. The surface of the fiber bundle is relatively rough. In addition, there are many gaps in the position of the fiber bundle criss-crossing and fiber criss-crossing, which is one of the reasons why the insulating pressboard can absorb water easily.

Figure 4c–j shows the SEM images of Z10-A30-P20, Z10-A30-P30, Z10-A60-P20, and Z10-A60-P30 with a magnification of 2000× and 20,000×. It is obvious that the surface structure of the insulating pressboard was changed by magnetron sputtering. A thin film was formed on the surface of the insulating paper plate by magnetron sputtering with different materials and different times. The surface of the treated sample was smoother and some gaps in the position of the fiber bundle crisscrossing and fiber crisscrossing were filled. On the surface of the cellulose the particles, with diameters of 100–200 nm, were distributed uniformly and compactly.

The chemical composition of the sample before, and after, treatment was analyzed by EDS. Figure 5a shows the molecular structure of cellulose. Cellulose is composed of Beta-d-pyran glucosyl group (glucose with loss of water). Therefore, the energy spectra of C, H, and O elements appear in the EDS images of the untreated insulating board (shown in Figure 5b). In contrast, the EDS analysis results of the selected area, in Figure 6c, are shown in Figure 6d. The characteristic peaks of Zn, F, and Al represent the existence of Zn, F, and Al.

#### 3.1.2. Cross-Section Topography Analysis

The observed SEM image of Z10-A60-P30 after being ion polished is shown in Figure 6. Figure 6a shows the cellulose and its surface films. It can be seen that the section of cellulose after wide beam argon ion polishing was relatively flat. The elements mapping images in Figure 6 shows that there were many Zn, Al, and F elements on the surface of the cellulose. Moreover, an obvious stratification phenomenon appeared. The Zn element was concentrated in the innermost layer, followed by Al element, and the F element was in the outermost layer. In addition, Zn, F, and Al were not only found on the surface of the insulating board, but were also distributed in small amounts on its inside surface.

From Figure 6a,b, the preparation and stratification of the ZnO-Al_2_O_3_-PTFE film was proved by a chemical aspect. The stratification phenomenon in Figure 6c is clearer. Three different substances were deposited on the surface of the insulating paper; three thin films, of varying thickness on the surface of the cellulose, can be seen in the amplification of the SEM. The bottom is cellulose, on top of the cellulose is ZnO film, next is the Al film, and the PTFE film is at the top. The thickness of the PTFE film (601 nm) is larger than that of the ZnO film (288 nm), and the Al_2_O_3_ film (72 nm) is only a very thin layer. Figure 6c,d demonstrated the preparation and stratification of the ZnO-Al_2_O_3_-PTFE film. To sum up, it can be stated that ZnO, Al_2_O_3_ and PTFE were successfully constructed on the surface of the cellulose insulating pressboard in order, thereby showing the structure of a three-layer nanometer film. The argon ion cross-section polishing had little influence on the thickness of the deposited film, therefore, the thickness of the films, after cross-section polishing, was reliable. The layer thickness mainly depended on the sputtering efficiency and time of the PTFE, ZnO, and Al_2_O_3_.

### 3.2. Hydrophobicity and Hygroscopicity Analysis

The insulating pressboard is composed of cellulose, hemicellulose, and lignin and contains a large number of hydroxyl groups, which easily form hydrogen bonds with water. In addition, the SEM results also show that cellulose presents a typical porous multilayer structure, which makes it easier to absorb water. However, when the insulation board absorbs water, its insulation performance and mechanical properties are sharply reduced, so if the insulation pressboard surface has a good hydrophobicity, then it can be very good at preventing water damage on the insulation pressboard. The contact angles of the prepared samples, to the water and mineral oil, are shown in Figure 7. An amount of 10 μL of liquid was dripped into each sample. Figure 8 shows the dynamic change of the liquid on the sample surface, describing the whole process from the liquid contact surface to the liquid entering the insulating board completely.

As can be seen from Figure 7, the contact angle between the water and oil of the untreated insulating pressboard was zero, which means that when water or oil touched the surface of the insulating pressboard, the liquid could enter the insulating board quickly, due to the hydrogen of the cellulose. In contrast, the contact angle, between the plated sample and water, was more than 110 degrees, which is hydrophobic. On the one hand, the film formed by the magnetron sputtering filled the gap on the original insulating board surface and reduced the capillarity of the insulating board. On the other hand, the outermost PTFE nanofilm had a greater C–F bond, which shielded the hydroxyl group of the inner cellulose and improved the hydrophobicity.

Figure 8a shows that the contact angle of a 30 min PTFE coating sample was higher than 20 min of coating. The sample hydrophobic angle of PTFE for 20 min was 112.5°, and 113.1°, respectively. The sample hydrophobic angle of PTFE for 30 min was 115.6°, and 117.45°, respectively. Water droplets could be maintained on the surface of the insulating pressboard for a long time, but with the increase of time, they gradually penetrated the insulating pressboard. Within 15–20 min, the contact angle of the sample surface to water was still greater than 90°. Finally, after about 35–40 min, the water completely penetrated the pressboard. It was apparent that the hydrophobic properties of the insulating paperboard increased as the thickness of the PTFE increased. At the same time, the Al_2_O_3_ film under the PTFE layer had a slight effect on the overall hydrophobic properties. The hydrophobic properties of the Al_2_O_3_ film deposited for a longer period of time were slightly higher.

In addition, the contact angles of different samples to the mineral insulating oil for the transformer were also tested, as shown in Figure 8b. It was found that the film prevented moisture and oil from entering the insulating board. The insulating oil, dripping onto the surface of the insulating board, will not enter the insulating board after an hour. This means that the insulating board requires a longer immersion process.

### 3.3. Measurement of Oil Immersion Rate

To evaluate the oil immersion rate of the treated, and untreated, samples, the thermogravimetry (TG) and the derivative thermogravimetry (DTG) curves of all the samples were measured. There were two peaks in the DTG curve of the oil-immersed insulating board, which indicates that the oil-immersed insulating board contains cracking and volatilization of two substances, during the heating process. The TG curve of the insulation pressboard can be roughly divided into three parts. As shown in Figure 9, the decomposition of the insulating oil occurred at about 100–200 °C, while the insulating paper decomposed at 250–380 °C. Therefore, the insulation board oil immersion rate can be obtained by a TG curve. The immersion rate can be calculated by dividing the weight of the oil by the total mass, as shown in Figure 10. For different insulating boards, after 48 h of the same immersion process, their immersion rates were similar. Combined with the test results of the contact angle (Figure 8b), we can infer that, although the film on the surface of the insulating pressboard slows down the process of oil entering the pressboard, after the normal immersion process, it can still achieve the same immersion rate as the normal pressboard.

### 3.4. Frequency Dielectric Properties of the Sputtered Pressboard

The dielectric constant and dielectric loss of the treated, and untreated, samples are shown in Figure 11. The dielectric constant increased in the low frequency range, while the dielectric loss increased at the same time. This was termed a low frequency dispersion (LFD) by Jonscher [37]. However, the dielectric constant of the coated pressboard was much higher than that of the untreated insulating pressboard in the low frequency region, and there were 1–2 loss peaks in Figure 11 of the dielectric loss tangent. The polarization reached equilibrium by overcoming a certain potential energy, so it needed to absorb a certain amount of energy, which is the reason for the dielectric loss peak formation. The establishment of polarization takes a certain amount of time, and the relaxation time constant of polarization can be calculated by the Debye model [38].
(1)$$\tan\delta =\frac{(\varepsilon_{s}-\varepsilon_{\infty })\omega \tau }{\varepsilon_{s}+\varepsilon_{\infty }\omega^{2}\tau^{2}}$$
where tanδ is the tangent of the dielectric loss angle, τ is the relaxation time, ω is the angular frequency, $\varepsilon_{s}$ is the static dielectric constant, and $\varepsilon_{\infty }$ is the optical frequency dielectric constant. When $\omega \tau =\sqrt{\frac{\varepsilon_{s}}{\varepsilon_{\infty }}}$, tanδ is at its maximum. Therefore, for the samples with ZnO, Al_2_O_3_, and PTFE films, the relaxation time, τ, is about 10^−1^–10^−5^ s. From the relaxation time, it can be judged that the loss peak is not caused by electronic displacement polarization or ion displacement polarization. In order to determine the cause of the polarization, insulating pressboards with different films were tested, and the results are shown in Figure 12. It can be clearly seen from Figure 12 that pressboards containing ZnO have a much higher dielectric constant than other pressboards. The loss peak appears in the frequency dielectric spectrum of the insulating pressboard with ZnO film, while the dielectric loss of the pressboard without the ZnO film is slightly lower than that of the untreated insulating paperboard. It can be preliminarily inferred that the polarization is caused by ZnO.

This phenomenon was also found in ZnO/LDPE composites, with a dielectric loss peak in nanometer ZnO/LDPE and no dielectric loss peak in micron ZnO/LDPE [39]. The authors think that the dielectric loss peak is caused by the addition of nanoparticles, which leads to the dipole moment formed by the interfacial polarization in the composite [39]. The loss peak also exists in the frequency dielectric spectrum of ZnO ceramics. The activation energy was calculated to determine the polarization type; the authors think that thermionic polarization caused by interstitial defects of Zn^2+^ ions and the oxygen vacancy defect is the reason for the loss peak [40]. However, this paper can judge that the cause of polarization is caused by ZnO; however, the root cause of this polarization still needs to be further studied.

From the standpoint of a single substance (as shown in Figure 13), Al_2_O_3_ and PTFE reduced the dielectric loss and dielectric constant. There were many dipoles on the surface of the insulating paperboard—the presence of the film may have reduced the surface dipoles. Furthermore, as the thickness of Al_2_O_3_ increases, the dielectric constant and the dielectric loss increase, and as the thickness of PTFE increases, the dielectric constant and the dielectric loss decrease.

### 3.5. AC Breakdown Performance of Different Samples

The breakdown strength is an important parameter to evaluate the dielectric insulation performance. For the untreated insulating pressboard and magnetron sputtering insulating pressboard, AC breakdown was carried out at a boost rate of 1 kV/s. The results of the AC breakdown experience are shown in Table 2. The breakdown voltage of oil-impregnated insulating pressboard was fitted with two-parameter Weibull distribution [41] (as shown in Figure 14), and the breakdown voltage with a failure probability of 63.2% was taken as the scale parameter to describe the breakdown performance of oil-impregnated insulating pressboard,
(2)$$F(t)=1-e^{-{\left(\frac{t}{\alpha }\right)}^{\beta }}$$
where F(t) is the failure probability, *t* is the number of experiments, *α* is the shape parameter, and β is the scale parameter. The shape parameter and the scale parameter of each sample are shown in Table 3. The breakdown field strength of the modified pressboard was larger than that of the untreated pressboard. It was calculated from Table 2 that the AC breakdown field strength of the modified samples was increased by 12–17%.

## 4. Conclusions

The SEM and EDS show that the nano-sized particles were attached on the pressboard surface, and the three-layer, nano-structured ZnO-Al_2_O_3_-PTFE functional film was successfully fabricated on the cellulose insulation pressboard surface by RF magnetron sputtering. In addition, the gap between the cellulose was shown to be filled with nanoparticles. The surface of the cellulose with high roughness became smooth. In the SEM image of the cross-section, it was demonstrated that the film presents an obvious stratification phenomenon.

From the analysis of the contact angles, the functional film shields the hydroxyl group of the inner cellulose and improves hydrophobicity. It takes a long time for moisture to get into the pressboard. At the same time, insulating oil enters the paper at a slower speed; however, the result of the TG shows that the film did not change the oil immersion rate of the insulating pressboard.

The FDS results show that ZnO introduces a new relaxation polarization into the insulating board. The frequency of the relaxation polarization is within the range of 10^0^–10^4^ Hz. The AC breakdown field strength of the modified samples increases by 12–17%, compared to the fresh sample.

By depositing a ZnO-Al_2_O_3_-PTFE, multifunctional composite film on the surface of insulating pressboard, this study provides a method to comprehensively improve the electrical properties of insulating paper and its ability to inhibit moisture.

## Figures and Tables

**Figure 1 polymers-11-01367-f001:**
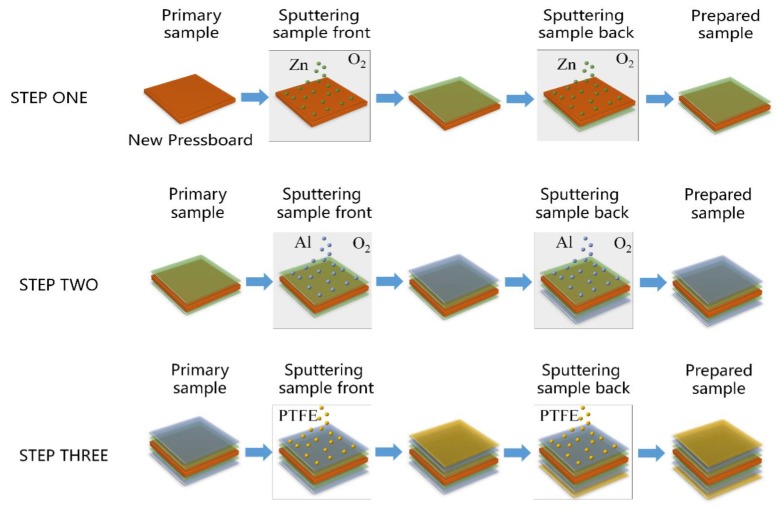
The process of magnetron sputtering of ZnO-Al_2_O_3_-PTFE multilayer films.

**Figure 2 polymers-11-01367-f002:**
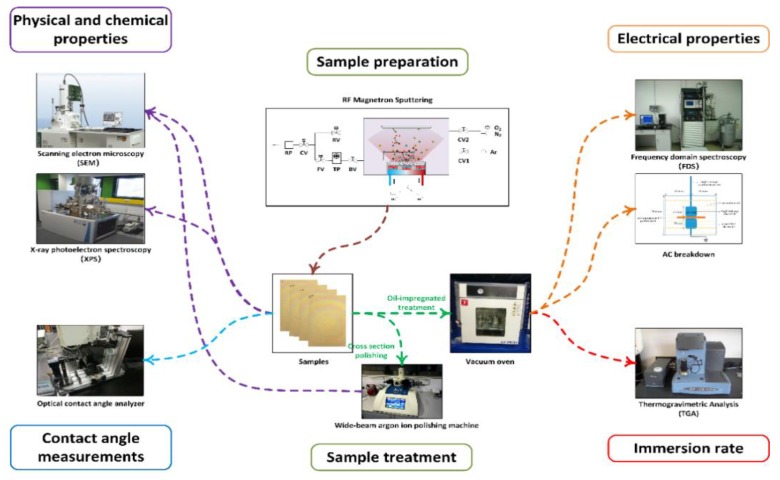
Flow chart of sample preparation and characterization.

**Figure 3 polymers-11-01367-f003:**
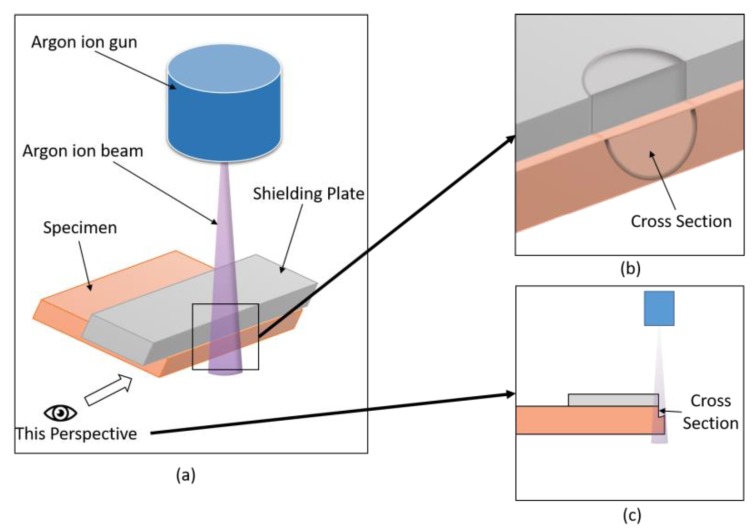
Schematic diagram of argon ion cross-section polishing (CP) process (**a**); partial enlarged detail of sample after argon ion cross-section polishing (**b**); and side view of argon ion cross-section polishing (**c**).

**Figure 4 polymers-11-01367-f004:**
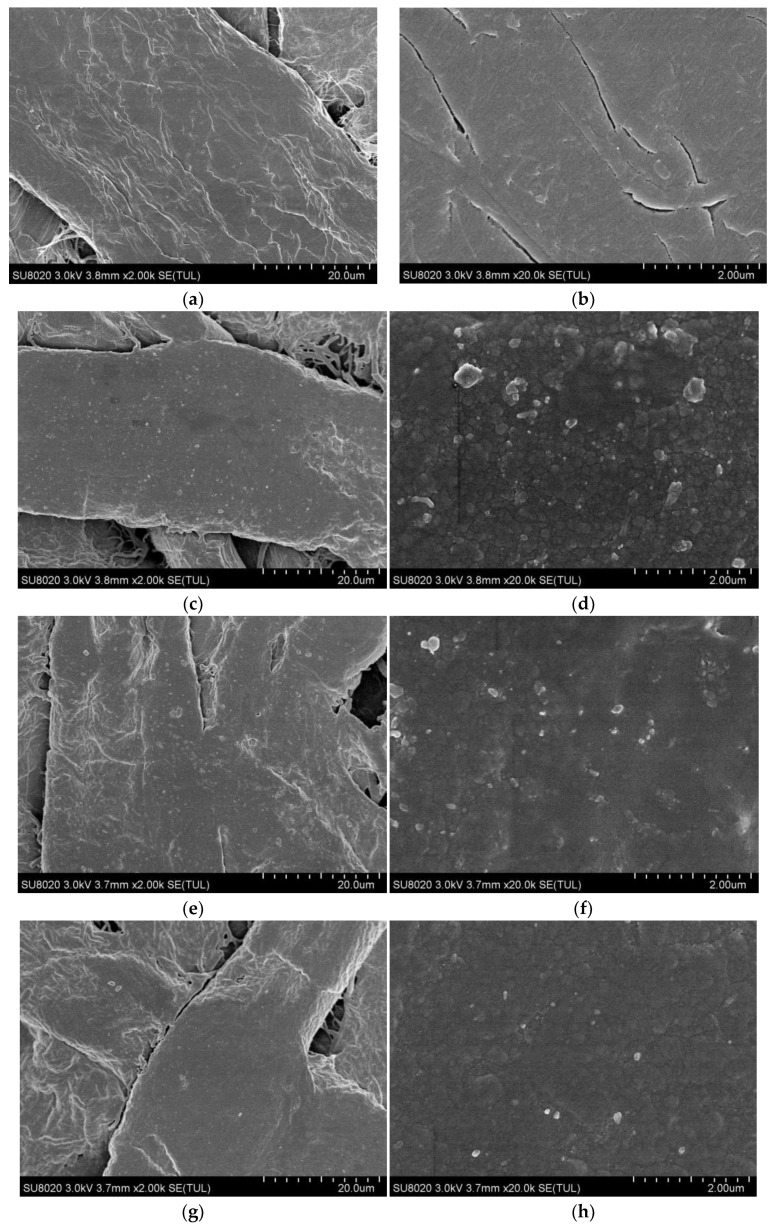
Scanning electron microscopy (SEM) images of the untreated insulating pressboard surface at 2k× magnification (**a**) and at 20k× magnification (**b**); surface of sample Z10-A30-P20 at 2k× magnification (**c**) and at 20k× magnification (**d**); surface of sample Z10-A30-P30 at 2k× magnification (**e**) and at 20k× magnification (**f**); surface of sample Z10-A60-P20 at 2k× magnification (**g**) and at 20k× magnification (**h**); and surface of sample Z10-A60-P30 at 2k× magnification (**i**) and at 20k× magnification (**j**).

**Figure 5 polymers-11-01367-f005:**
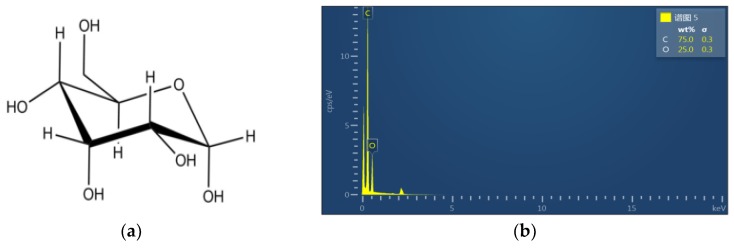
Molecular formula of cellulose (**a**) and EDS of the untreated pressboard (**b**).

**Figure 6 polymers-11-01367-f006:**
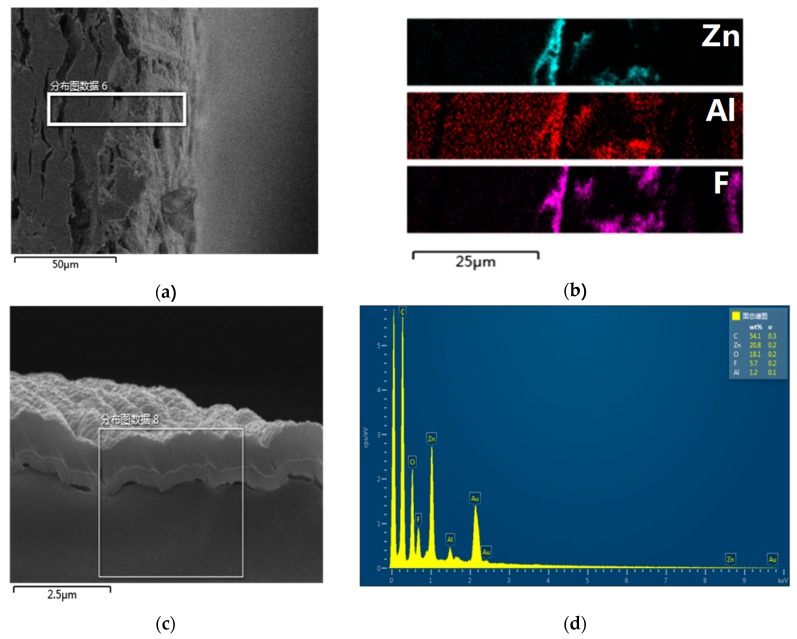
Scanning electron microscopy (SEM) image of sample Z10-A60-P30 after cross-section polishing at 1k× magnification (**a**) and element mapping of the selected area (**b**); scanning electron microscopy (SEM) image of sample Z10-A60-P30 at 15k× magnification, (**c**) and energy dispersive spectrometry (EDS) of the selected area (**d**).

**Figure 7 polymers-11-01367-f007:**
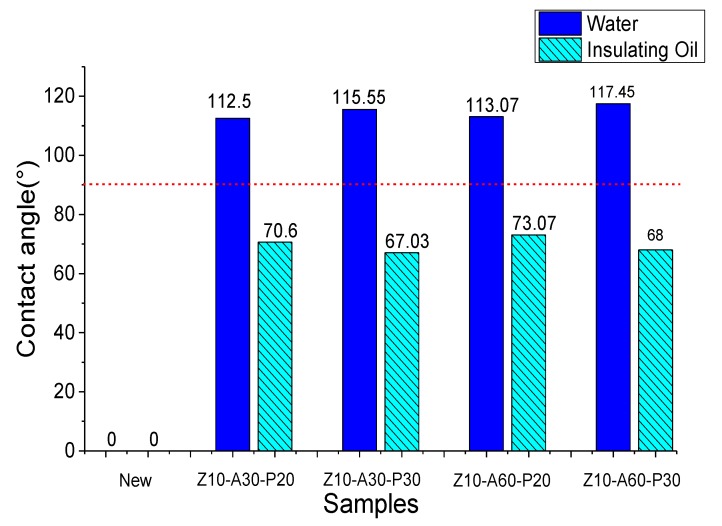
Contact angle of untreated insulating pressboard and modified insulating pressboard to water and mineral oil.

**Figure 8 polymers-11-01367-f008:**
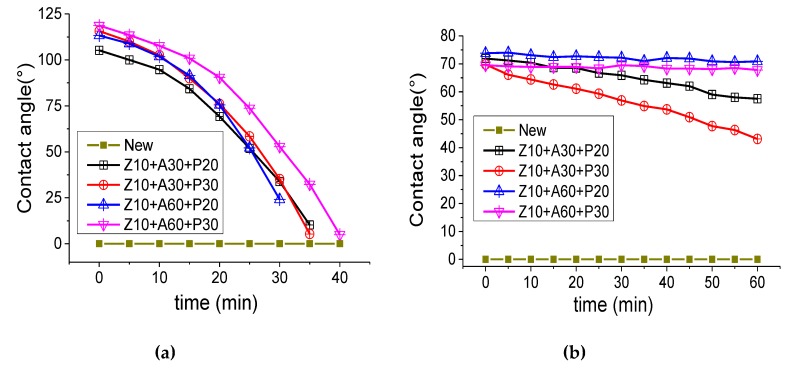
Dynamic change of contact angle of untreated insulating board and modified insulating board to water (**a**) and mineral oil (**b**).

**Figure 9 polymers-11-01367-f009:**
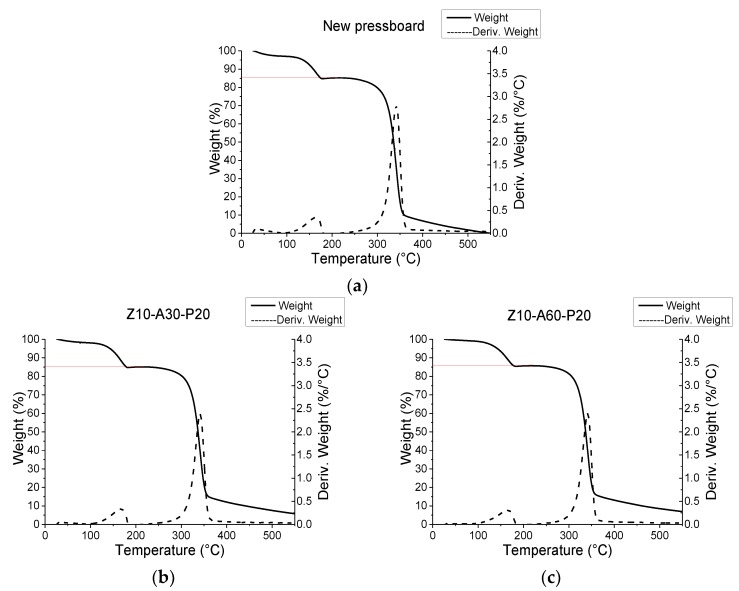
The thermogravimetry (TG) and the derivative thermogravimetry (DTG) curves of the untreated pressboard (**a**); sample Z10-A30-P20 (**b**); sample Z10-A60-P20 (**c**); sample Z10-A30-P30 (**d**); and sample Z10–A60–P30 (**e**).

**Figure 10 polymers-11-01367-f010:**
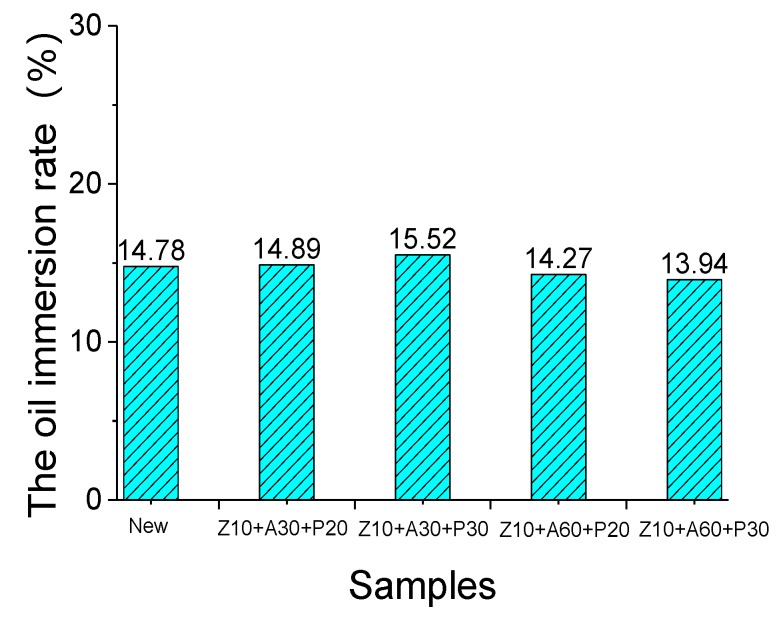
The immersion rate of the untreated insulating pressboard and modified insulating pressboard.

**Figure 11 polymers-11-01367-f011:**
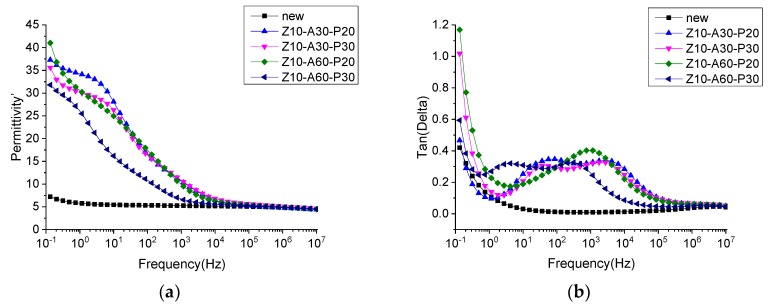
Frequency dielectric spectrum (**a**) and dielectric loss spectra (**b**) of the untreated insulating pressboard and modified insulating pressboard.

**Figure 12 polymers-11-01367-f012:**
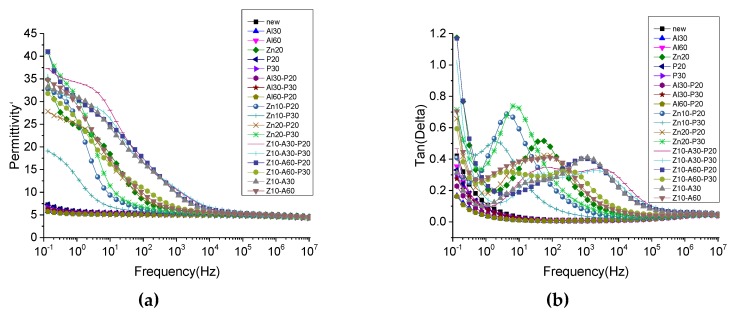
Frequency dielectric spectrum (**a**) and dielectric loss spectra (**b**) of the insulating pressboard with different films.

**Figure 13 polymers-11-01367-f013:**
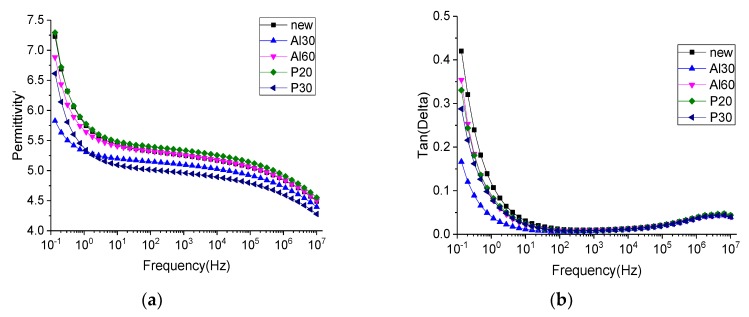
Frequency dielectric spectrum (**a**) and dielectric loss spectra (**b**) of the insulating pressboard with single film.

**Figure 14 polymers-11-01367-f014:**
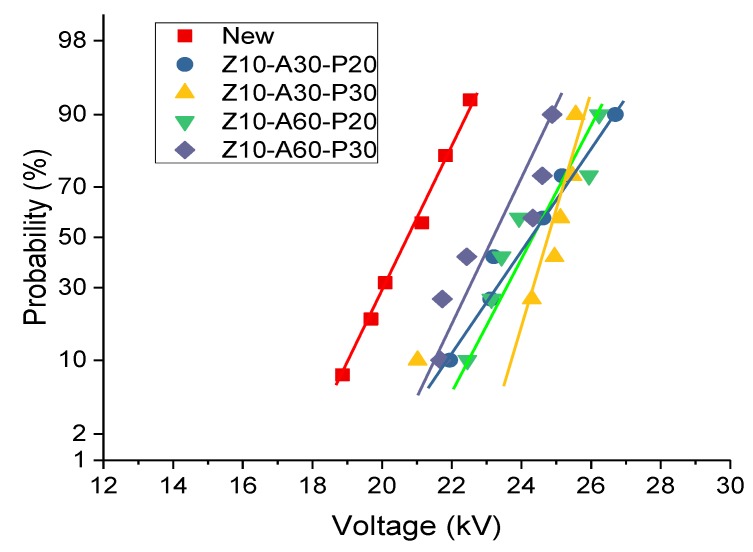
The two-parameter Weibull distribution of AC breakdown voltage.

**Table 1 polymers-11-01367-t001:** Sample preparation parameters.

Sample Name	Time for Sputtering ZnO	Time for Sputtering Al_2_O_3_	Time for Sputtering PTFE
New	0	0	0
Z10-A30-P20	10 min	30 min	20 min
Z10-A30-P30	10 min	30 min	30 min
Z10-A60-P20	10 min	60 min	20 min
Z10-A60-P30	10 min	60 min	30 min

**Table 2 polymers-11-01367-t002:** AC breakdown voltage of the untreated and modified insulating pressboards.

Test Number	AC Breakdown Voltage (kV)				
	Untreated Pressboard	Z10-A30-P20	Z10-A30-P30	Z10-A60-P20	Z10-A60-P30
1	19.68	24.62	24.3	23.14	24.88
2	20.09	23.21	24.95	23.93	21.68
3	22.53	21.94	21.02	22.45	24.6
4	18.86	25.17	25.12	26.23	22.43
5	21.14	23.12	25.47	23.43	21.73
6	21.82	26.7	25.56	25.94	24.33

**Table 3 polymers-11-01367-t003:** Scale parameter and shape parameter of two-parameter Weibull distribution.

Sample	Scale Parameter α (kV/mm)	Shape Parameter β
New	42.569	18.410
Z10-A30-P20	49.748	16.368
Z10-A30-P30	50.018	27.463
Z10-A60-P20	49.754	18.106
Z10-A60-P30	47.828	20.479

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
