# Peer review of "Fabrication of ZnO-Al2O3-PTFE Multilayer Nano-Structured Functional Film on Cellulose Insulation Polymer Surface and Its Effect on Moisture Inhibition and Dielectric Properties"

_polymers, 2019, doi:10.3390/polym11081367_

Round 1
Reviewer 1 Report
In this manuscript, Liu et al. report on an exhaustive characterization of the morphological, compositional and electrical properties of a multilayered nanostructure film used to increase the breakdown field and the resistance to humidity of cellulose polymers used in power transformers.
The novelty of the work is mainly the simultaneous exploitation of PTFE, mainly used to inhibit moisture, and of ZnO/alumina coatings, which are conversely effective in improving electrical insulation.
The paper deserves publication in Polymers journal after minor revision.
However, as a general comment, English should be mandatorily improved in some sections of the paper. Some parts, as the abstract, are very clear and well-written, whereas some others require necessarily proofreading by a native speaker, because they may result quite hard to understand (e.g. lines 36-39, or lines 209-212).
Herewith a detailed list of comments and suggestions:
Line 19. Expand PTFE acronym.
Lines 36-39. Authors should better highlight the role of oil immersion in cellulose insulation polymer-based transformers.
Line 70. Authors forgot to mention that their work aims at “improving” the electrical properties.
Line 77. Replace “Al2O3 thin film” with “multilayer thin film”, because sputtering has been used for all the layers and not only for alumina.
Lines 81-83. I find the description of magnetron sputtering technique rather useless. Sputtering is indeed a well-known thin film deposition technique, and does not need to be introduced for the n-th time in a scientific paper. Moreover, there is an error in the description, because magnetic field does not act on the Argon ions, but on the electrons, by confining their movement near the toroid and leading to a higher plasma density and consequently to a higher deposition rate. For the same reason, I would eliminate Fig.1 (or at least eliminate the mechanical scheme of pumps and valves). I suggest to write down just the experimental details of the different sputtering depositions (base pressure, Ar pressure, oxygen flow in case of reactive sputtering of ZnO and alumina, power, and so on).
Line 93. Just for clarity, authors should briefly recall here the reason why the sequence ZnO/Al2O3/PTFE has been chosen for their complex coating, highlighting the role each single layer has in improving the pressboard properties.
Line 124. It’s not clear why Argon ion polishing is performed. It should be included in the text.
Figures 4c-j. Are the SEM images of coated pressboards taken before or after the ion polishing? It should be specified.
Figure 5. The term “new” is used here (and somewhere else in the text) to identify the original pressboard. I would avoid to use “new”, it’s more clear “untreated”, or “uncoated”.
Line 193. A PTFE layer of about 600 nm is claimed, even after ion polishing. Is PTFE layer thicker than the others (alumina + ZnO) to account for the subsequent removal of material caused by ion polishing? Please clarify.
Lines 284-286. It appears clear from fig. 12 that only films containing ZnO show higher permittivity and peaks in the dielectric loss spectrum. Therefore, I’d just write “It can be clearly seen from figure 12 that pressboards containing ZnO have much higher dielectric constant than other pressboards”, thus highlighting that the presence of alumina or PTFE (alone or as a bilayer) seems to have influence neither on the permittivity nor on the dielectric loss.
Line 290. Expand LDPE acronym.
Author Response
Herewith a detailed list of comments and suggestions:
Line 19. Expand PTFE acronym.
Response:Thanks. It has been supplemented. Please refer line 19.
Lines 36-39. Authors should better highlight the role of oil immersion in cellulose insulation polymer-based transformers.
Response:Thanks. The role of oil immersion was added in line 36-39.
“The cellulose insulation polymer itself has poor heat dissipation, and the air gap in the fiber reduces its dielectric strength, so the cellulose insulating polymer needs to be immersed in the insulating oil. The insulation oil fills the air gap, play a role of heat dissipation, sealing and insulation coordination”.
Line 70. Authors forgot to mention that their work aims at “improving” the electrical properties.
Response:Thanks. It has been supplemented in line 88-90.
Line 77. Replace “Al2O3thin film” with “multilayer thin film”, because sputtering has been used for all the layers and not only for alumina.
Response:Thanks. Corrected. Line 97.
Lines 81-83. I find the description of magnetron sputtering technique rather useless. Sputtering is indeed a well-known thin film deposition technique, and does not need to be introduced for the n-th time in a scientific paper. Moreover, there is an error in the description, because magnetic field does not act on the Argon ions, but on the electrons, by confining their movement near the toroid and leading to a higher plasma density and consequently to a higher deposition rate. For the same reason, I would eliminate Fig.1 (or at least eliminate the mechanical scheme of pumps and valves). I suggest to write down just the experimental details of the different sputtering depositions (base pressure, Ar pressure, oxygen flow in case of reactive sputtering of ZnO and alumina, power, and so on).
Response:Thanks. The description of magnetron sputtering technique has been deleted. The Fig.1 has been deleted. The experimental details of the different sputtering depositions, such as base pressure, Ar pressure, oxygen flow in case of reactive sputtering of ZnO and alumina, power, and so on was described.
Please refer line 97-104.
Line 93. Just for clarity, authors should briefly recall here the reason why the sequence ZnO/Al2O3/PTFE has been chosen for their complex coating, highlighting the role each single layer has in improving the pressboard properties.
Response:Thanks. The reason why the sequence ZnO/Al2O3/PTFE, and the role of each single layer has been added in line 70-82.
In recent years, wide band-gap semiconductor material zinc oxide (ZnO) has attracted more and more attention [1], with advantages of high electron mobility, nonlinear resistance [1-3]. Nano-ZnO is believed to be beneficial to charge transport [3], Its resistance decreases with the increase of the electric field, which will help balance the electric field [4]. Low concentration doped nano-ZnO in LDPE will be beneficial to reduce the internal electric field distortion and improve its breakdown performance [4-5].
Takahashi, K.; Yoshikawa, A.; Sandhu, A. Wide Bandgap Semiconductors, Fundamental Properties and Modern Photonic and Electronic Devices, Wide Bandgap Semiconductors: Fundamental Properties and Modern Photonic and Electronic Devices, Gupta T K; Carlson W G. A grain-boundary defect model for instability/stability of a ZnO varistor, Journal of Materials Science, 1985, 20, 3487-3500. Ellmer K; Klein A; Rech B. Transparent Conductive Zinc Oxide, Springer Berlin Heidelberg, Wang, X.; Cheng, X.; Chen, S. Q.; Zheng, X. Q.; Tu, D. M. Dielectric properties of the composites of nano-zno/low-density polyethylene, Zhongguo Dianji Gongcheng Xuebao/Proceedings of the Chinese Society of Electrical Engineering, Cheng, Y.; Bai, L.; Yu, G.; Zhang, X. Effect of Particles Size on Dielectric Properties of Nano-ZnO/LDPE Composites, Materials,
Al2O3 is a famous insulating material, which is often used as coating or nano-filler for insulating paper [6-10]. Results show that the nano-Al2O3 doped insulation paper possesses the better dielectric properties and AC breakdown strength [9]. And the Al2O3 functional film constructed on the surface of insulating board can realize binding charge and effectively inhibit charge injection from electrode [10].
Yan, S.; Liao, R.; Yang, L.; Zhao, X.; He, L. Influence of nano-Al2O3 on electrical properties of insulation paper under thermal aging, 2016 IEEE International Conference on High Voltage Engineering and Application (ICHVE), 2016, 1-4. Bobzin, K.; Lugscheider, E.; Maes, M.; Piñero, C. Relation of hardness and oxygen flow of Al2O3 coatings deposited by reactive bipolar pulsed magnetron sputtering, Thin Solid Films, 2006, 494, 255–262. Beysens, D.; Chatain, D.; Evesque, P.; Garrabos, Y. Synthesis of -Al2O3 thin films using reactive high-power impulse magnetron sputtering, Europhysics Letters, 2006, 3, 36002–36005. Liu, H.; Chi, M.; Chen, Q.; Gao, Z.; Wei, X. Analysis of dielectric characteristics of nano-al2o3 modified insulation pressboard, Zhongguo Dianji Gongcheng Xuebao/Proceedings of the Chinese Society of Electrical Engineering, 2017, 37, 4246-4253. Jian, H.; Yanqing, L.; Ruijin, L.; Guoyong, L.; Qiang, L.; Chao, T. Fabrication of Al2O3 nano-structure functional film on a cellulose insulation polymer surface and its space charge suppression effect, Polymers, 2017, 9, 502.
PTFE has good insulation properties, corrosion resistance and hydrophobic properties. PTFE film can be used as the outermost protective layer of multi-layer film. It prevents moisture in oil from entering into insulating pressboard [11-13], and on the other hand, it can protect the functional layer of inner layer from being damaged [14].
Jian, H.; Cong, L.; Yanqing, L.; Ruijin, L.; Qiang, L.; Chao, T. Preparation nano-structure polytetrafluoroethylene (PTFE) functional film on the cellulose insulation polymer and its effect on the breakdown voltage and hydrophobicity properties, Materials, 2018, 11, 851. Yong, J.; Fang, Y.; Chen, F.; Huo, J.; Yang, Q.; Bian, H.; et al. Femtosecond laser ablated durable superhydrophobic ptfe films with micro-through-holes for oil/water separation: separating oil from water and corrosive solutions, Applied Surface Science, 2016, 389, 1148-1155. Hou, W.; Wang, Q. Stable polytetrafluoroethylene superhydrophobic surface with lotus-leaf structure, Journal of Colloid & Interface Science, 2009, 333, 400-403. Wan, Y.; Yu, Y.; Cao, L.; Zhang, M.; Gao, J.; Qi, C. Corrosion and tribological performance of PTFE-coated electroless nickel boron coatings, Surface and Coatings Technology, 2016, 307, 316-323.
Therefore, the PTFE layer was deposited on the outmost layer, the middle layer is Al2O3, the innermost layer is ZnO.
Line 124. It’s not clear why Argon ion polishing is performed. It should be included in the text.
Response: Thanks. The original expression is not clear enough, we have supplemented the Argon ion polishing in line 133-139.
In this paper, we only did cross-section polishing (Gatan 697 llion II, Gatan, America) to obtain a cross section of the multilayer deposited film for SEM/EDS analysis. The principle of argon ion section polishing is shown in the figure below. The purpose of argon ion cross section polishing is to obtain a cross section of the sample. Figure a is the schematic diagram of argon ion polishing, figure b is the effect of sample section polishing, and figure c is the side view. As the processing energy source, Argon ion beam hits perpendicular to the surface of a specimen and the side of the sample is exposed to ion beam with the help of baffle. Argon ions generated bombarded the sample surface at a high speed. And under one hour and 50 minutes at 6.5 keV of ion beam energy, 20 minutes at 3 keV, and 20 minutes at 1 keV, ions made a cross section obtained perpendicular to the surface of a specimen.
Figure. Schematic diagram of argon ion cross-section polishing (CP) process (a); partial enlarged detail of sample after argon ion cross-section polishing (b) and side view of argon ion cross-section polishing (c).
Figures 4c-j. Are the SEM images of coated pressboards taken before or after the ion polishing? It should be specified.
Response:Thanks. The SEM images of coated pressboards shown in Figures 4c-j was not being ion polishing. This has been specified in the revised paper. Please refer line 160-161.
Figures 7a-d the SEM images of the deposited multilayer film is after being cross-section polishing. It was also specified in line 199. Please refer line 199.
Figure 5. The term “new” is used here (and somewhere else in the text) to identify the original pressboard. I would avoid to use “new”, it’s more clear “untreated”, or “uncoated”.
Response:Thanks. Only “untreated” was used in the revised paper.
Line 193. A PTFE layer of about 600 nm is claimed, even after ion polishing. Is PTFE layer thicker than the others (alumina + ZnO) to account for the subsequent removal of material caused by ion polishing? Please clarify.
Response:Thanks. The argon ion cross-section polishing has little influence on the thickness of deposited film, the thickness of films after section polishing is reliable. The layer thickness mainly depends on the sputtering efficiency and time of PTFE, ZnO and Al2O3.
Please refer line 216-218.
Lines 284-286. It appears clear from fig. 12 that only films containing ZnO show higher permittivity and peaks in the dielectric loss spectrum. Therefore, I’d just write “It can be clearly seen from figure 12 that pressboards containing ZnO have much higher dielectric constant than other pressboards”, thus highlighting that the presence of alumina or PTFE (alone or as a bilayer) seems to have influence neither on the permittivity nor on the dielectric loss.
Response:Thanks. Corrected. Please refer line 310-311.
Line 290. Expand LDPE acronym.
Response:Thanks. Corrected. LDPE (low density plyethylene).
13) However, as a general comment, English should be mandatorily improved in some sections of the paper. Some parts, as the abstract, are very clear and well-written, whereas some others require necessarily proofreading by a native speaker, because they may result quite hard to understand (e.g. lines 36-39, or lines 209-212).
Response:Thanks. The revision paper has been language edited by the MDPI English editing service.

Reviewer 2 Report
Report on the manuscript:
Fabrication ZnO-Al2O3-PTFE Multilayer Nano-structured Functional Film on Cellulose Insulation Polymer Surface and Its Effect on Moisture Inhibition and Dielectric Property
By:
Cong Liu, Jian Hao, Yanqing Li and Ruijin Liao
Manuscript Number: Polymers-566346
In this work, the authors used RF magnetron sputtering to deposit ZnO-Al2O3-PTFE multilayer films on 0.45 mm thick insulating paper pressboard. They kept the deposition time of ZnO film to 10 minutes and varied that of Al2O3 (30 and 60 minutes) and PTFE (20 and 30 minutes) films.
The aim of the study is to improve the performance of the cellulose-insulating polymer in terms of its dielectric properties and its behavior with respect moisture. For that purpose, the authors used several characterization techniques: scanning electron microscopy coupled with an energy dispersive spectrometer, contact angle measurements, thermogravimetry and frequency dielectric spectroscopy.
Although this study is rather consistent, the paper would gain in quality with a little more discussion about the obtained properties of the samples. Nevertheless, the paper requires some improvements and answers to the following remarks before being accepted for publication.
1°/ In comparison with a similar study published by the same authors (Reference 23: Jian, H.; Cong, L.; Yanqing, L.; Ruijin, L.; Qiang, L.; Chao, T. Preparation nano-structure polytetrafluoroethylene (PTFE) functional film on the cellulose insulation polymer and its effect on the breakdown voltage and hydrophobicity properties, Materials, 2018, 11, 851), the pressboard paper, in this study, is covered by 2 additional films (ZnO and Al2O3).
a) The authors should begin by explaining the interest of these 2 additional films. and why they kept a constant thickness of the ZnO layer.
b) They should also explain why they chose such thickness of ZnO and why they kept it constant.
2°/ The obtained results show that the physical and physicochemical properties of the samples (hydrophobicity and hygroscopicity, immersion rate and dielectric properties) depend on the thickness of both Al2O3 and PTFE. Therefore, the authors must explain the role of each material in these properties. Obviously, they tried to do it for dielectric measurements (Fig. 12) but in a very confused way.
3°/ As the authors mentioned, “the breakdown voltage is an important parameter to evaluate the dielectric insulation performance”. Consequently,
a) The AC breakdown experiment must be well-described and the measured parameters well-defined;
b) The raw measured data must be presented (before those of Fig. 13) to prove the increase in AC breakdown field strength announced in the abstract of the paper.
c) The parameter F(t) of Eq. (1) must be defined;
4°/ Some technical deficiencies to be addressed:
a) The acronyms PTFE (polytetrafluoroethylene) must be defined at the beginning of the paper.
b) A reference is needed for the expression of tand.
c) All equations must be numbered (line 278: Eq. giving tand).
d) Several sentences in the text are wobbly and some grammar mistakes must be corrected.
e) Some references to format or to complete:
Ref 4: Liu, J.F.; Fan, X.H.; Zheng, H.B.; Zhang, Y.Y.; et.al. Aging condition assessment of transformer oil-immersed cellulosic insulation based upon the average activation energy method. Cellulose, 2019, 26(6):, 3891-3908.
Ref 16: Liang, N.; Liao, R.; Min, X.; Yang, M.; Yuan, Y. Influence of amine compounds on the thermal stability of paper-oil insulation. Polymers, 2018, 10, 891.
Author Response
g accepted for publication.
In comparison with a similar study published by the same authors (Reference 23: Jian, H.; Cong, L.; Yanqing, L.; Ruijin, L.; Qiang, L.; Chao, T. Preparation nano-structure polytetrafluoroethylene (PTFE) functional film on the cellulose insulation polymer and its effect on the breakdown voltage and hydrophobicity properties, Materials, 2018, 11, 851), the pressboard paper, in this study, is covered by 2 additional films (ZnO and Al2O3). a) The authors should begin by explaining the interest of these 2 additional films.
Response:Thanks. The reason why the sequence ZnO/Al2O3/PTFE has been chosen is added in line 70-83.
Al2O3 is a famous insulating material, which is often used as coating or nano-filler for insulating paper [1-5]. Results show that the nano-Al2O3 doped insulation paper possesses the better dielectric properties and AC breakdown strength [4]. And the Al2O3 functional film constructed on the surface of insulating board can realize binding charge and effectively inhibit charge injection from electrode [5].
Yan, S.; Liao, R.; Yang, L.; Zhao, X.; He, L. Influence of nano-Al2O3 on electrical properties of insulation paper under thermal aging, 2016 IEEE International Conference on High Voltage Engineering and Application (ICHVE), 2016, 1-4. Bobzin, K.; Lugscheider, E.; Maes, M.; Piñero, C. Relation of hardness and oxygen flow of Al2O3 coatings deposited by reactive bipolar pulsed magnetron sputtering, Thin Solid Films, 2006, 494, 255–262. Beysens, D.; Chatain, D.; Evesque, P.; Garrabos, Y. Synthesis of -Al2O3 thin films using reactive high-power impulse magnetron sputtering, Europhysics Letters, 2006, 3, 36002–36005. Liu, H.; Chi, M.; Chen, Q.; Gao, Z.; Wei, X. Analysis of dielectric characteristics of nano-al2o3 modified insulation pressboard, Zhongguo Dianji Gongcheng Xuebao/Proceedings of the Chinese Society of Electrical Engineering, 2017, 37, 4246-4253. Jian, H.; Yanqing, L.; Ruijin, L.; Guoyong, L.; Qiang, L.; Chao, T. Fabrication of Al2O3 nano-structure functional film on a cellulose insulation polymer surface and its space charge suppression effect, Polymers, 2017, 9, 502.
PTFE has good insulation properties, corrosion resistance and hydrophobic properties. PTFE film can be used as the outermost protective layer of multi-layer film. On the one hand, it prevents moisture in oil from entering into insulating pressboard [6-8], and on the other hand, it can protect the functional layer of inner layer from being damaged [9].
Jian, H.; Cong, L.; Yanqing, L.; Ruijin, L.; Qiang, L.; Chao, T. Preparation nano-structure polytetrafluoroethylene (PTFE) functional film on the cellulose insulation polymer and its effect on the breakdown voltage and hydrophobicity properties, Materials, 2018, 11, 851. Yong, J.; Fang, Y.; Chen, F.; Huo, J.; Yang, Q.; Bian, H.; et al. Femtosecond laser ablated durable superhydrophobic ptfe films with micro-through-holes for oil/water separation: separating oil from water and corrosive solutions, Applied Surface Science, 2016, 389, 1148-1155. Hou, W.; Wang, Q. Stable polytetrafluoroethylene superhydrophobic surface with lotus-leaf structure, Journal of Colloid & Interface Science, 2009, 333, 400-403. Wan, Y.; Yu, Y.; Cao, L.; Zhang, M.; Gao, J.; Qi, C. Corrosion and tribological performance of PTFE-coated electroless nickel boron coatings, Surface and Coatings Technology, 2016, 307, 316-323.
b) They should also explain why they chose such thickness of ZnO and why they kept it constant.
Response:Thanks. The reason for keeping thickness of ZnO constant was supplemented. Please refer line 70-75.
In recent years, wide band-gap semiconductor material zinc oxide (ZnO) has attracted more and more attention [10], with advantages of high electron mobility, nonlinear resistance [10-12]. Nano-ZnO is believed to be beneficial to charge transport [12], Its resistance decreases with the increase of the electric field, which will help balance the electric field [13]. Low concentration doped nano-ZnO in LDPE will be beneficial to reduce the internal electric field distortion and improve its breakdown performance [13-14].
Our experiments with ZnO are shown below:
ZnO films with different sputtering times were prepared on the surface of pressboards by radio frequency magnetron sputtering, and the basic electrical properties of ZnO films were tested as shown in the Figure 1-4. Because ZnO films are semiconductor oxides, their resistivity is lower than that of ordinary insulators.It is quite understandable that his presence reduces the volume resistivity of insulating board, and as the coating time increases, its volume resistivity decreases more severely (as shown in Figure 1).
Similarly, it can be seen from Figure 2 that ZnO film sharply reduces the surface resistivity of insulating pressboard, and the longer the coating time is, the more severe the decrease will be.This undoubtedly increases the probability of insulation pressboard flashover along the surface. Therefore, the ZnO layer cannot appear on the surface, and in this paper, it is the innermost layer.
The DC breakdown (The boost rate is 1kV/s) and pre-stress DC breakdown tests (Add the electric field 16 kV/mm for 5 minutes, and then boost the pressure at the rate of 1kV/s) are also carried out, as shown in Figure 3-4. It can be seen that the thin film sputtering ZnO for 10 minutes has positive significance for the improvement of DC breakdown field strength and pre-stress breakdown field strength. However, the DC breakdown field strength of the sample ZnO20 is less than that of ZnO10, and the pre-pressed breakdown field strength is even lower than that of untreated insulating board. It means that the overall performance reduction caused by the low insulation performance of ZnO material is greater than the improvement caused by the functional film.
In conclusion, the low surface resistivity of ZnO film makes it unable to be used as the outermost film, so in the composite film, it can be used as the innermost layer. In addition, the comprehensive comparison of resistivity, dc breakdown field strength and preloading breakdown field strength shows that the samples sputtered ZnO for 10 minutes are better than those sputtered ZnO for 20 minutes. Therefore, 10 minutes of ZnO film was selected in the innermost layer of the composite film.
Figure 1. Volume resistivity of samples sputtered ZnO films with different time
Figure 2. Surface resistivity of samples sputtered ZnO films with different time
Figure 3. DC breakdown field strength of samples sputtered ZnO films
Figure 4. Pre-pressure breakdown field strength of samples sputtered ZnO films
Takahashi, K.; Yoshikawa, A.; Sandhu, A. Wide Bandgap Semiconductors, Fundamental Properties and Modern Photonic and Electronic Devices, Wide Bandgap Semiconductors: Fundamental Properties and Modern Photonic and Electronic Devices, Gupta T K; Carlson W G. A grain-boundary defect model for instability/stability of a ZnO varistor, Journal of Materials Science, 1985, 20, 3487-3500. Ellmer K; Klein A; Rech B. Transparent Conductive Zinc Oxide, Springer Berlin Heidelberg, Wang, X.; Cheng, X.; Chen, S. Q.; Zheng, X. Q.; Tu, D. M. Dielectric properties of the composites of nano-ZnO/low-density polyethylene, Zhongguo Dianji Gongcheng Xuebao/Proceedings of the Chinese Society of Electrical Engineering, Cheng, Y.; Bai, L.; Yu, G.; Zhang, X. Effect of Particles Size on Dielectric Properties of Nano-ZnO/LDPE Composites, Materials,
The obtained results show that the physical and physicochemical properties of the samples (hydrophobicity and hygroscopicity, immersion rate and dielectric properties) depend on the thickness of both Al2O3and PTFE. Therefore, the authors must explain the role of each material in these properties. Obviously, they tried to do it for dielectric measurements (Fig. 12) but in a very confused way.
Response: Thanks. The role of each material in these properties is added in line 262-265 and 327-331.
As can be seen from Figures 8 and 9, it is apparent that the hydrophobic properties of the insulating paperboard increase as the thickness of the PTFE increases. At the same time, the Al2O3 film under the PTFE layer has a slight effect on the overall hydrophobic properties. The hydrophobic properties of the Al2O3 film deposited for a longer period of time are slightly higher.
From the Figure 10 and 11, TG shows that the films did not change the oil immersion rate of insulating pressboard.
ZnO increases the dielectric constant and dielectric loss and causes a dielectric loss peak, so the dielectric constant and dielectric loss images in the three-layer film are confusing (Fig. 12). However, from the standpoint of a single substance (in order to make the expression clearer, Figure 14 is added in line ), Al2O3 and PTFE reduce the dielectric loss and dielectric constant (there are many dipoles on the surface of the insulating paperboard, which may be the presence of the film to reduce the surface Dipole), and as the thickness of Al2O3 increases, the dielectric constant and dielectric loss increase, and as the thickness of PTFE increases, the dielectric constant and dielectric loss decrease.
As the authors mentioned, “the breakdown voltage is an important parameter to evaluate the dielectric insulation performance”. Consequently, a) The AC breakdown experiment must be well-described and the measured parameters well-defined;
Response: Thanks. The description of AC breakdown experiment was added. Please refer line 154-158.
b) The raw measured data must be presented (before those of Fig. 13) to prove the increase in AC breakdown field strength announced in the abstract of the paper.
Response: Thanks. The raw measured data was added in Table 2. Please refer line 358.
c) The parameter F(t) of Eq. (1) must be defined;
Response: Thanks. The expression of equation was added.
Please refer line 353-354.
Some technical deficiencies to be addressed: a) The acronyms PTFE (polytetrafluoroethylene) must be defined at the beginning of the paper.
Response: Thanks. Corrected. Please refer line 19.
b) A reference is needed for the expression of tan
Response: Thanks. The expression and reference of equation was added in line 306-309.
c) All equations must be numbered (line 278: Eq. giving tand).
Response: Thanks. The equation was numbered. Please refer line 307.
d) Several sentences in the text are wobbly and some grammar mistakes must be corrected.
Response: Thanks. Some errors have been corrected.
Please refer line 242-245, 281, 314-316, 323.
e) Some references to format or to complete:
Response: Thanks. Format of References 4, 6,12,13,16,25,26,30 has been changed.
Please refer line 391-450。
Ref 4: Liu, J.F.; Fan, X.H.; Zheng, H.B.; Zhang, Y.Y.; et.al. Aging condition assessment of transformer oil-immersed cellulosic insulation based upon the average activation energy method. Cellulose, 2019, 26(6):, 3891-3908.
Response: Thanks. Corrected. Please refer line 391-393.
Ref 16: Liang, N.; Liao, R.; Min, X.; Yang, M.; Yuan, Y. Influence of amine compounds on the thermal stability of paper-oil insulation. Polymers, 2018, 10, 891.
Response: Thanks. Corrected. Please refer line 419-420.
Thanks. The revision paper has been language edited by the MDPI English editing service.

Round 2
Reviewer 2 Report
Report on the manuscript:
Fabrication of ZnO-Al2O3-PTFE Multilayer Nano-structured Functional Film on Cellulose Insulation Polymer Surface and Its Effect on Moisture Inhibition and Dielectric Properties
By:
Cong Liu, Jian Hao, Yanqing Li and Ruijin Liao
Manuscript Number: Polymers-566346-v2
The authors have more or less provided answers to my remarks. I, therefore, consider that the new version can now be published as it is. However, there are still some technical deficiencies that need to be corrected:
1°/ The acronym LDPE (low density polyethylene) must be first defined on line 74 and not on line 319.
2°/ In Table 2, the units of time and breakdown strength must be indicated.
3°/ Line 310 : When ωτ = √εs/ε∞, d(tanδ)/dω is equal to 0, tanδ is at its maximum.
4°/ References for books or book chapters must be completed in accordance with the requirements of the journal.
Author Response
The authors have more or less provided answers to my remarks. I, therefore, consider that the new version can now be published as it is. However, there are still some technical deficiencies that need to be corrected:
1°/ The acronym LDPE (low density polyethylene) must be first defined on line 74 and not on line 319.
Response: Thanks. Corrected. Please refer in line 74. Red color.
2°/ In Table 2, the units of time and breakdown strength must be indicated.
Response: Thanks. Corrected. Please refer in line 359. Red color.
3°/ Line 310 : When ωτ = √εs/ε∞, d(tanδ)/dω is equal to 0, tanδ is at its maximum.
Response: Thanks. Corrected. Please refer in line 310. Red color.
4°/ References for books or book chapters must be completed in accordance with the requirements of the journal.
Response: Thanks. Corrected. Please refer in line 387, 433, 474. Red color.